# Protective Effect of Exercise Against Depression, Anxiety, and Stress Among University Students Based on Their Level of Academic Performance

**DOI:** 10.3390/medicina60101706

**Published:** 2024-10-17

**Authors:** Ibrahim M. Gosadi

**Affiliations:** Department of Family and Community Medicine, Faculty of Medicine, Jazan University, P.O. Box 2349, Jazan 82621, Saudi Arabia; gossady@hotmail.com

**Keywords:** exercise, depression, anxiety, stress, students, academic performance

## Abstract

*Background and Objectives:* Certain lifestyle behaviors can act as either buffering or aggravating factors influencing the mental well-being of university students. The current investigation assessed the association between academic performance and the risk of anxiety, depression, and stress among university students and how physical activity might buffer the levels of these conditions. *Materials and Methods*: A cross-sectional design was utilized to reach a sample of university students from Saudi Arabia. Data about the demographics of the students, GPA, exercise regularity, and levels of anxiety, depression, and stress were collected. Logistic regression was performed to investigate the influence of exercise regularity on the presence of anxiety, depression, and stress, dividing the sample according to the GPA category. *Result*: A total of 506 students were recruited. The mean age of the students was 22 years, and 53% of the students were men. An assessment of the levels of depression, anxiety, and stress indicated that 60%, 57%, and 40% have abnormal levels of these conditions, respectively. Students with a GPA of “good” or lower exhibited higher levels of stress. Exercising regularly was associated with a protective effect against depression and stress (*p* < 0.05). However, the buffering effect of exercise was more apparent among students with a GPA of “very good” or above compared to students with lower academic achievement. *Conclusions*: The lack of a statistically significant association between physical activity regularity and mental well-being among students classified as low academic achievers may indicate the need for additional psychological support in addition to the promotion of a physically active lifestyle.

## 1. Introduction

Mental well-being can be influenced by individual characteristics and modified by several social factors. The internal characteristics can be related to age, gender, and genetic risk [1]. However, some social factors might be associated with an impact on mental well-being. For example, exposure to certain social or occupational contexts might expose some individuals to a higher risk of anxiety, depression, and stress.

One of the social determinants that has been indicated to influence mental well-being is the academic environment’s impact on students’ mental health. There is growing evidence suggesting that students can be exposed to mental health problems during their studies. In an Italian study involving 3002 students aged between 15 and 16, it was concluded that one-fifth of the students exhibited mental health problems, with the risk associated with being a victim of bullying, smoking, alcohol consumption, and a negative school environment [2]. Furthermore, in a Turkish study involving a sample of 1617 university students, it was reported that the prevalence of abnormal levels of depression, anxiety, and stress were 27%, 47%, and 27%, respectively. These abnormalities were more prevalent among women, students in their early university years, and those who were not satisfied with their scores [3]. Furthermore, in another Chinese study involving a sample of 600 students where first-year students were compared to fifth-year students, it was concluded that the average scores of depression, anxiety, and stress were more among fifth-year students in comparison to first-year students. This indicates the importance of addressing mental well-being among university students, especially during later years [4].

The importance of assessing mental well-being among university students stems from the notion that impaired mental health can have an impact on the academic performance of students. Furthermore, the academic stress itself is also a predictor of mental distress among the students. For example, in a Japanese study involving a sample of 1823 university students, it was concluded that students suffering from impaired mental health in the first semester of their studies were at higher risk of poor academic performance during the remaining study period [5]. This notion suggests that poor mental health can harm overall academic performance. However, in another Chinese study that involved a sample of 1804 secondary-school students, it was concluded that academic stress can lead to mental distress, where the occurrence of mental distress was higher among girls in comparison to boys [6]. This indicates the complexity of association between the academic environment on mental health of students, where poor mental well-being can influence academic performance, and academic distress, which can be considered as a predictor of mental health status.

Worsening mental health among university students has been indicated to be reduced by several buffering factors. In a systematic review involving 31 studies that assessed factors associated with mental health among UK university students, it was concluded that psychological-strength characteristics, such as coping strategies and emotional intelligence, supportive parents, positive self-image, and the ability to recognize the development of mental health problems and seek healthcare, were associated with a reduced risk of mental health problems among the students [7]. Similarly, in a recent study that assessed mental well-being among university students, it was conceptualized that perceived stress induced by academic, financial, and family pressures is modified by factors related to student characteristics such as loneliness, self-esteem, personality, and coping strategies [8]. Furthermore, personality traits have also been indicated as factors affecting mental well-being of students [9,10].

There is growing evidence indicating that certain lifestyle characteristics can have a buffering effect on mental health among university students. In a longitudinal Spanish study that followed 1292 students to assess the development of depression over a year, it was concluded that healthy sleeping, physical activity, and social support were associated with lower odds of depression [11].

Furthermore, in a study that assessed the presence of psychiatric disorders among a sample of 147 Tunisian university students, it was concluded that physical activity was associated with a protective effect among the students [12]. Finally, in a study that recruited a sample of 472 university students from Saudi Arabia to assess the association between the presence of depression, anxiety, and stress and walking, it was concluded that walking was associated with reduced levels of anxiety and depression among male students, but not female students [13].

The current literature assessing levels of anxiety, depression, and stress among university students indicates that the academic environment, including the mental distress associated with the level of academic performance, can increase the risk of developing these conditions. Furthermore, certain lifestyle characteristics, such as physical activity, can act as buffering factors, enhancing the mental well-being of the students. However, the levels of anxiety, depression, and stress according to academic performance and physical activity levels are not well understood. The current investigation aims to assess the association between academic performance and the risk of anxiety, depression, and stress among university students from Saudi Arabia. Furthermore, an analysis of how physical activity might buffer levels of anxiety, depression, and stress according to the level of academic performance is also carried out.

## 2. Method

### 2.1. Study Context

This paper presents the findings from a cross-sectional assessment among university students from Jazan, in the southwest of Saudi Arabia. The recruitment of participants took place between February and March 2023. Data collection was performed online. Ethical approval to perform the study was secured from the Standing Committee of Scientific Research at Jazan University (approval number: REC-44/04/365, dated 14 November 2022). An information letter was shared with the students explaining the objectives of the study, the targeted population, and the nature of the assessment. The information letter was followed by a consent form; students who consented to participate were allowed to access the questionnaire, while those who refused were excluded. No identifying information was gathered from the students, and all participation was anonymous. The students had the right to reject participation or withdraw at any phase of the recruitment.

### 2.2. Data Collection Tool

Data presented in the current investigation was collected through two main components. The first component was a demographic section that included items related to age, gender, history of diagnosis with a chronic disease, history of smoking and khat chewing, social status, and current GPA. Students were also asked about their exercise behavior by assessing their weekly exercise habits. The second component assessed levels of anxiety, depression, and stress using the DASS 21 questionnaire, whose development, validity, reliability, and scoring was conducted by Antony et al. [14]. The assessment of validity and reliability of the DASS 21 questionnaire by Antony et al. revealed high internal consistency, where Cronbach’s Alpha levels for depression, anxiety, and stress were 0.94, 0.87, and 0.91, respectively. Additionally, concurrent validity of the DASS 21 questionnaire was performed by comparing the findings of the DASS 21 to the Beck Depression Inventory, Beck Anxiety Inventory, and State-Trait Anxiety Inventory, revealing positive correlations varying between 0.51 and 0.85 between the measured parameters. Furthermore, an Arabic version of the DASS 21 questionnaire was utilized in the current investigation to suit the language preferences of the targeted students [15]. In addition to the established validity and reliability of the DASS 21 questionnaire in previous research, the estimation of the Cronbach’s Alpha test for the current data reached 0.95.

### 2.3. Data Collection Process

To facilitate reaching the students, the questionnaire was converted to an electronic format using Google Forms. A web link was created and advertised on social media platforms associated with university students, such as WhatsApp and Telegram. Students who were registered to the university at the time of recruitment were included. Individuals who were graduates of the university or not registered at the university during the recruitment period were excluded.

Convenience sampling was used to reach the targeted students. This method of sampling was utilized since it was difficult to establish a sampling frame when targeting the students via social media platforms. A sample size of 500 university students was estimated based on the findings of a previous study that assessed levels of anxiety, depression, and stress among students from the same university [13]. This sample size was calculated using the StatCal function of Epi Info software (Version 7.2.3.0). It was sufficient to detect a prevalence of anxiety, depression, and stress of 50% with a 95% confidence level and a 5% error margin, with an additional 30% increase to account for excluded cases not meeting the inclusion criteria.

### 2.4. Data Analysis

Data were analyzed using the Statistical Package for the Social Sciences software (version 25). All variables presented in the current analysis were either binary or categorical, except for age. Frequencies and proportions were used to summarize binary and categorical variables. Age was initially summarized via mean and standard deviation (SD). Distribution of the age was assessed via visualization of age distribution using a histogram and normal distribution curve. To allow performing subsequent inferential analysis, age was categorized as a binary variable using the mean as a cut-off point where the students were classified as either less than 22, or 22 years old or more.

Evaluations of the presence of depression, anxiety, and stress were performed according to the scoring system described elsewhere [14]. Students were classified as either having normal or abnormal levels of anxiety, depression, and stress. Students who had a final score of 9 or less within the depression scale were classified as normal while those who scored more than 9 were classified as having an abnormal depression level. Students who had a final score of 7 or less within the anxiety scale were classified as having normal levels of anxiety while those who scored more than 7 were labeled as having abnormal levels of anxiety. A stress final score of 14 or less was classified as normal levels of stress, and students who had a final score of 15 or more were classified as having abnormal stress levels.

Additionally, students were categorized based on their GPA into three groups: those with a GPA of “good” or lower, “very good”, and “excellent”. This classification was made according to the university grading system where students who reported a GPA of 4.5/5 or more were classified as having an excellent GPA, students who reported having a GPA ranging between 3.75–<4.5/5 were classified as having a very good GPA, and those with a GPA lower than 3.75/5 were classified as students with a good or lower GPA. Finally, the level of exercise was classified based on the number of days per week spent performing any type of physical activity. Due to the low levels of physical activity in the recruited sample, students were grouped into either those who exercised at least once per week or those who did not exercise at all during the week.

Inferential statistics were performed in two steps. First, the chi-squared test was used to assess statistical differences in the demographics of the sample, their exercise frequency, and levels of anxiety, depression, and stress according to their GPA category and exercise frequency. Second, to investigate the influence of exercise frequency on the presence of anxiety, depression, and stress, univariate logistic regression was conducted, with the sample divided into three groups based on GPA category. This was followed by multivariate logistic regression to account for the possible confounding effects of other demographic factors on the association between exercise frequency and levels of anxiety, depression, and stress. A *p* value of 0.05 or less was considered statistically significant.

## 3. Results

A total of 506 students met the inclusion criteria and completed the data collection. Forty-three cases were excluded for not meeting the inclusion criteria. The mean age of the students was 22 (SD: 2.1) years, with 53% being men and 66% coming from health-related colleges. The majority of the sample had no diagnosis of chronic disease, were non-smokers, had never chewed khat, and were single. When asked about their weekly exercise frequency, only 37% reported exercising at least once per week, while the remaining 63% reported not engaging in any form of weekly exercise.

Table 1 displays the distribution of the sample demographics according to GPA category. The proportion of students with higher GPAs was greater among younger students, women, those not diagnosed with any chronic disease, and those with no history of smoking or khat chewing (*p* < 0.05). Exercise frequency, social status, and field of study did not show statistically significant differences according to GPA level.

An assessment of levels of anxiety, depression, and stress among the recruited students indicates a high prevalence of abnormalities. A total of 60% of the students were classified as having abnormal levels of depression, 57% as having abnormal levels of anxiety, and 40% as having abnormal levels of stress. The distribution of anxiety, depression, and stress levels according to GPA is illustrated in Table 2. Only the distribution of stress levels was statistically significant according to GPA, suggesting that students with a GPA of “good” or lower were more likely to exhibit higher stress levels compared to those in other GPA categories.

Table 3 displays the distribution of levels of anxiety, depression, and stress according to weekly exercise frequency. Statistically significant associations were identified between exercise frequency and levels of depression and stress, with marginal significance for levels of anxiety. In all conditions, the proportion of students who reported engaging in at least one weekly exercise session was lower among those classified as having abnormal levels of anxiety, depression, and stress. This suggests that exercise frequency may be associated with a buffering effect that enhances students’ mental well-being.

Odds ratios for anxiety, depression, and stress based on exercise frequency as a dependent variable are displayed in Table 4. The analysis by GPA category indicates that the protective effect of exercise frequency varies according to GPA. Among students classified as having a GPA of “very good”, the odds ratios show a protective effect against anxiety, depression, and stress (*p* < 0.05). However, among students classified as having a GPA of “excellent”, only the protective effect against stress was statistically significant (*p* < 0.005). For students classified as having a GPA of “good” or lower, no statistically significant associations were detected regarding the impact of exercise frequency on anxiety, depression, and stress.

Table 5 summarizes the influence of various variables on anxiety, depression, and stress, with a particular emphasis on exercise and gender, while controlling for age, history of chronic disease, smoking, and khat chewing. The influence of gender appears to be more pronounced among students classified as having a GPA of “excellent” and “good” or lower (*p* < 0.05). However, the protective effect of exercise frequency remained statistically significant among students classified as having a GPA of “very good” across all conditions, and in reducing stress levels among those classified as having a GPA of “excellent”, even when controlling for other factors (*p* < 0.05).

## 4. Discussion

The current study identified a high proportion of university students classified as having abnormal levels of anxiety, depression, and stress. According to the findings, academic performance is associated with stress levels, with students classified as having a GPA of “good” or lower exhibiting higher levels of stress compared to students with a GPA of “very good” or higher. An assessment of the associations between exercise frequency and abnormalities in anxiety, depression, and stress across the whole sample suggests that regular exercise is associated with a protective effect against depression and stress. However, when examining the association between exercise frequency and these abnormalities while classifying the sample according to GPA level, it was noted that regular exercise has a protective effect against all conditions among students classified as having a “very good” GPA, and a protective effect against stress among students classified as having an “excellent” GPA. This protective effect of exercise remained statistically significant even when controlling for other variables such as age, gender, history of chronic diseases, smoking, and khat chewing.

The findings of the current investigation can be compared to similar studies. A Malaysian study involving a sample of 506 university students concluded that high levels of anxiety, depression, and stress were present, with severity levels primarily associated with age and gender [16]. Another study involving a sample of 332 university students from the United Arab Emirates reported higher scores for depression, anxiety, and stress among female students [17].

This is similar to the findings of the current study, which also indicate higher levels of abnormalities in these conditions and highlight the significant impact of gender. However, a Chinese study involving a sample of 1648 university students from Hong Kong found that gender and age were not associated with depression, anxiety, and stress. Instead, a stronger influence of the economic situation and living alone was linked to these conditions [18].

Additionally, a Bangladeshi study involving a sample of 359 medical students found no association between academic performance and levels of anxiety, stress, and depression. However, it was noted that these conditions were more prevalent among students attending public medical colleges compared to those in private medical colleges [19]. This suggests the potential impact of the economic situation and academic environment on mental well-being. Although the current study did not assess the impact of the economic situation, these variations indicate that there may be differences between populations when addressing the mental well-being of university students.

In a Chinese study by Zhang et al., which involved a sample of 600 university students and assessed levels of depression, anxiety, and stress in relation to academic performance, it was concluded that high levels of stress, depression, and anxiety were associated with lower academic performance, with stress having a greater effect compared to the other conditions [4]. The current study detected a statistically significant association between stress and academic performance, but not between anxiety and depression. This suggests that academic performance among university students may be more strongly associated with stress levels compared to anxiety and depression.

This finding was similar to those of a Lebanese study by Kharroubi et al., which involved a sample of 261 university students. This study identified a higher proportion of students suffering from stress (75%) compared to those suffering from depression and anxiety (42% and 36%, respectively). Kharroubi et al. also found an association between levels of anxiety, depression, and stress and academic performance, concluding that students with low-to-moderate GPAs were more likely to experience higher stress levels compared to students with high GPAs [20]. These findings align with the current investigation, which detects statistically significant differences in stress levels according to academic performance.

In another Egyptian study by Fawzy and Hamed, which involved a sample of 700 university students from a medical school, it was reported that levels of depression, anxiety, and stress exceeded 60% of the sample, with lower academic achievement being one of the factors contributing to higher scores of these conditions [21]. Additionally, their multivariate analysis highlighted the strong impact of gender, finding that women were at higher risk of developing abnormal scores for these conditions, which was similar to the findings of the current study.

In addition to the identified association between academic performance and mental well-being, satisfaction with the education system can be another contributing factor. A Lebanese study involving a sample of 1617 university students concluded that students who were satisfied with their education had lower levels of anxiety, depression, and stress [3]. Although the current study does not assess satisfaction with the education system, this represents another important element that should be considered when evaluating mental well-being among university students.

Studies assessing the association between an active physical lifestyle and mental well-being among university students are limited. The study by Roldan-Espinola et al. concluded that maintaining a healthy lifestyle, including physical activity, was associated with a lower risk of developing major depressive disorder [11]. Additionally, the study by Ghali et al. found a statistically significant association between physical activity and a lower risk of psychiatric disorders [12]. Finally, the study by Bahri et al. concluded that walking was associated with reduced levels of anxiety among male university students [13]. These findings, along with those of the current study, highlight the important role of an active lifestyle in enhancing the mental well-being of university students. Promoting an active lifestyle among university students can be recommended as a strategy to improve mental well-being.

### 4.1. Practical Implications

The multiple levels of analysis conducted in the current study indicate associations between academic performance, physical activity, and mental well-being. First, students classified as lower academic achievers experience higher levels of stress compared to others. Second, regular exercise can have a protective impact on mental well-being among university students, with this impact being more pronounced among students with a “very good” or “excellent” GPA. The lack of a statistically significant association between physical activity and mental well-being among students classified as low academic achievers may suggest the need for additional psychological interventions, rather than solely promoting a healthy and physically active lifestyle.

The current literature investigating interventions to enhance mental well-being and academic performance among university students shows limited evidence, and there is a need for more robust studies. A meta-analysis of randomized controlled trials involving 2428 university students reported a small and non-significant effect of electronic health interventions on reducing alcohol and tobacco use, as well as on mood and anxiety [22]. Another systematic review and meta-analysis of 17 trials assessing the impact of web-based and computer-delivered cognitive behavioral therapy on anxiety, depression, and stress found that these interventions could be effective in improving mental health compared to no intervention. However, the included studies were noted to have a moderate risk of bias [23].

A separate systematic review and meta-analysis of 26 randomized controlled trials concluded that the effects of mental health interventions on university students’ well-being could last between 3 and 12 months post-intervention, suggesting a longer-lasting impact on depression and anxiety compared to stress levels [24].

### 4.2. Limitations

The current study has several strengths and weaknesses. A key strength is the ability to reach a heterogeneous sample of university students with varying degrees of academic performance and demographic factors. This diversity enabled a comprehensive analysis of the impact of physical activity on mental well-being across different levels of academic achievement.

However, a major limitation of the study is the low level of physical activity reported among participants, which restricted the ability to further analyze the impact of different types and intensities of physical activity on mental well-being. This limitation is consistent with the existing literature indicating low physical activity levels among Saudis, particularly among female students [25]. Additionally, the findings are limited by the impact of other psychological factors which were not investigated in the current study, such as emotional intelligence, the presence of supportive parents, positive self-image, and mental healthcare-seeking behavior which might require conducting future research to indicate how physical activity interacts with these factors. Finally, this study utilized a cross-sectional design to assess the association, and assessment of temporality is not possible and it is not clear whether mental distress led to an impact on academic performance or whether the academic performance led to mental distress. However, the findings of the current study, along with the relevant literature on factors influencing mental well-being among university students, underscore the importance of implementing organizational programs for the early detection of at-risk individuals and the development of suitable preventive health interventions based on risk assessments.

## 5. Conclusions and Future Research

The current study found a protective impact of regular exercise on reducing levels of depression and stress among university students. However, the effect of exercise on mental well-being appears to be more pronounced in students with higher GPAs compared to those with lower academic achievement. This suggests that other influencing factors may be at play for students with lower GPAs. Future research should explore the interactions between specific demographic factors and lifestyle characteristics, and how these interactions impact students’ overall mental well-being. Additionally, the findings highlight the importance of considering students’ GPA, lifestyle, and demographic characteristics when assessing those at risk of mental health deterioration and designing appropriate interventions based on these determinants.

## Figures and Tables

**Table 1 medicina-60-01706-t001:** Distribution of demographic characteristics among a sample of 506 university students from Jazan, Saudi Arabia, according to GPA *.

	Good or Less	Very Good	Excellent	Total	*p* Value
Age					<0.001
Less than 22	41 [30.1%]	72 [43.6%]	121 [59%]	234 [46.2%]	
22 or more	95 [69.9%]	93 [56.4%]	84 [41%]	272 [53.8%]	
Gender					<0.001
Male	99 [72.8%]	90 [54.5%]	80 [39%]	269 [53.2%]	
Female	37 [27.2%]	75 [45.5%]	125 [61%]	237 [46.8%]	
Diagnosis with a chronic disease					
None	111 [81.6%]	139 [84.2%]	190 [92.7%]	440 [87%]	0.005
Yes	25 [18.4%]	26 [15.8%]	15 [7.3%]	66 [13%]	
Smoking					<0.001
Never	90 [66.2%]	140 [86.4%]	170 [88.5%]	400 [81.6%]	
Current or previous	46 [33.8%]	22 [13.6%]	22 [11.5%]	90 [18.4%]	
Khat chewing					<0.001
Never	110 [80.9%]	155 [95.7%]	185 [96.4%]	450 [91.8%]	
Current or previous	26 [19.1%]	7 [4.3%]	7 [3.65]	40 [8.2%]	
Social status					0.352
Single	124 [91.2%]	149 [90.3%]	193 [94.1%]	466 [92.1%]	
Married/Divorced/Widowed	12 [8.8%]	16 [9.7%]	12 [5.9%]	40 [7.9%]	
Specialty					0.266
Health	82 [60%]	109 [66.1%]	142 [69.3%]	333 65.9%]	
Art/Science	53 [39.3%]	56 [33.9%]	63 [30.7%]	172 [34.1%]	
Exercise					0.836
Once per week or more	53 [39%]	59 [35.8%]	74 [36.1%]	186 [36.8%]	
Less than once per week	83 [61%]	106 [64.2%]	131 [63.9%]	320 [63.2%]	

* GPA categorization: students were categorized based on their GPA into three groups: those with a GPA of “good” or lower, “very good”, and “excellent”. This classification was made according to the university grading system where students who reported a GPA of 4.5/5 or more were classified as having an excellent GPA, students who reported having a GPA ranging between 3.75–<4.5/5 were classified as having a very good GPA, and those with a GPA lower than 3.75/5 were classified as students with a good or lower GPA.

**Table 2 medicina-60-01706-t002:** Distribution of levels of anxiety, depression, and stress among a sample of 506 university students from Jazan, Saudi Arabia, according to GPA *.

	Good or Less	Very Good	Excellent	Total	*p* Value
Anxiety					0.805
Normal	56 [41.2%]	74 [44.8%]	90 [43.9%]	220 [43.5%]	
Abnormal	80 [58.8%]	91 [55.2%]	115 [56.1%]	286 [56.5%]	
Depression					0.359
Normal	49 [36%]	72 [43.6%]	78 [38%]	199 [39.3%]	
Abnormal	87 [64%]	93 [56.4%]	127 [62%]	307 [60.7%]	
Stress					0.027
Normal	69 [50.7%]	107 [64.8%]	129 [62.9%]	305 [60.3%]	
Abnormal	67 [49.3%]	58 [35.2%]	76 [37.1%]	201 [39.6%]	

* GPA categorization: students were categorized based on their GPA into three groups: those with a GPA of “good” or lower, “very good”, and “excellent”. This classification was made according to the university grading system where students who reported a GPA of 4.5/5 or more were classified as having an excellent GPA, students who reported having a GPA ranging between 3.75–<4.5/5 were classified as having a very good GPA, and those with a GPA lower than 3.75/5 were classified as students with a good or lower GPA.

**Table 3 medicina-60-01706-t003:** Distribution of levels of anxiety, depression, and stress among a sample of 506 university students from Jazan, Saudi Arabia, according to exercising frequency.

Condition	Exercising Regularly at Least Once a Week		Total	*p* Value
	Yes	No		
Anxiety				0.063
Normal	91 [41.4%]	129 [58.6%]	220 [100%]	
Abnormal	95 [33.2%]	191 [66.3%]	286 [100%]	
Depression				0.01
Normal	86 [43.2%]	113 [56.85]	199 [100%]	
Abnormal	100 [32.6%]	207 [67.4%]	307 [100%]	
Stress				<0.001
Normal	131 [43%]	174 [57%]	305 [100%]	
Abnormal	55 [27.4%]	146 [72.6%]	201 [100%]	

**Table 4 medicina-60-01706-t004:** Univariate regression analysis of anxiety, depression, and stress based on exercise frequency among a sample of 506 university students from Jazan, Saudi Arabia, according to GPA *.

Dependent Variable	Group	Outcome	Odds Ratio	95% CI	*p* Value
Exercising once or more per week [reference group: Not exercising on weekly basis]	Good or less	Anxiety	1.1	0.5–2.2	0.76
		Depression	0.7	0.8–1.5	0.48
		Stress	0.9	0.4–1.9	0.96
	Very good	Anxiety	0.4	0.2–0.8	0.015
		Depression	0.4	0.2–0.7	0.007
		Stress	0.2	0.1 -0.6	0.001
	Excellent	Anxiety	0.4	0.4–1.3	0.30
		Depression	0.7	0.4–1.3	0.39
		Stress	0.4	0.2–0.7	0.005

* GPA categorization: students were categorized based on their GPA into three groups: those with a GPA of “good” or lower, “very good”, and “excellent”. This classification was made according to the university grading system where students who reported a GPA of 4.5/5 or more were classified as having an excellent GPA, students who reported having a GPA ranging between 3.75–<4.5/5 were classified as having a very good GPA, and those with a GPA lower than 3.75/5 were classified as students with a good or lower GPA.

**Table 5 medicina-60-01706-t005:** Multivariate regression analysis of anxiety, depression, and stress based on exercise frequency and gender among a sample of 506 university students from Jazan, Saudi Arabia, according to GPA +.

Group	Dependent Variables ++	Anxiety: OR [95% CI]	Depression: OR [95% CI]	Stress: OR [95% CI]
Good or less	Exercise	1.1 [0.5–2.4]	0.7 [0.3–1.4]	0.9 [0.4–2.0]
	Gender	0.4 [0.1–0.9] *	0.7 [0.3–1.8]	0.4 [0.1–0.9] *
Very good	Exercise	0.4 [0.2–0.9] *	0.4 [0.1–0.8] *	0.3 [0.1–0.7] *
	Gender	0.2 [0.1–0.5] **	0.5 [0.2–1.1]	2.0 [0.9–4]
Excellent	Exercise	0.9 [0.5–1.8]	1.0 [0.4–1.5]	0.4 [0.2–0.9] *
	Gender	0.2 [0.1–1.9] **	0.2 [0.1–0.5] **	0.3 [0.1–0.7] *

+ GPA categorization: students were categorized based on their GPA into three groups: those with a GPA of “good” or lower, “very good”, and “excellent”. This classification was made according to the university grading system where students who reported a GPA of 4.5/5 or more were classified as having an excellent GPA, students who reported having a GPA ranging between 3.75–<4.5/5 were classified as having a very good GPA, and those with a GPA lower than 3.75/5 were classified as students with a good or lower GPS. ++ controlling for age, history of a chronic disease, smoking, and khat chewing within each group. * *p* value < 0.05, ** *p* value < 0.001.

## Data Availability

Dataset available on request from the authors.

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
