# Peer review of "Protective Effect of Exercise Against Depression, Anxiety, and Stress Among University Students Based on Their Level of Academic Performance"

_medicina, 2024, doi:10.3390/medicina60101706_

Round 1
Reviewer 1 Report
Comments and Suggestions for Authors
Thank you for the opportunity to review this manuscript. This is an interesting research topic.
The authors described clearly about the steps they did in this study. But some crucial information needs to be clarified in this study to make it easier for the readers to understand. The researchers divided the sample according to the GPA category, but it is not clear what specific criteria they used in making the three GPA categories. In several tables (table 1, table 2, table 4) the researchers did analysis according to GPA categories, but the GPA categories were not shown anywhere in the tables. The researchers categorize the depression, anxiety, and stress level as abnormal and normal levels, but they did not state the specific criteria they used to make these categorizations. These are very crucial information and the readers need to have this information to understand the study appropriately.
Here is some more feedback on specific paragraphs in different sections.
In the introduction section, the authors used “education system’s impact” in paragraph 2. “Education system’s impact” is very broad. Can you narrow it down and make it more specific what impact factors are you focusing on?
On page 2 the paragraph on the association of academic environment and students’ mental well-being is confusing. The topic sentence said academic performance was an influencing factor for mental wellbeing. However, the study cited said the mental health problems had a negative impact on the academic performance in that study. They are two different topics. Please make it consistent what you want to state here.
Paragraph 2 in Page 2 mentioned multiple buffering factors based on a previous study, but none of these factors are included in this study. Please make it clear how does this paragraph fit in the introduction section.
Terms like “University environment” “lifestyle characteristics” are all very broad terms. Can you use more specific term to make the statement closely related to the research topic of this study?
In the method section, as DASS 21 is the main assessment the researchers used to collect information on students’ anxiety, depression, and stress, it is necessary to describe DASS 21 questionnaire’s validity briefly in this manuscript.
It is good you make it clear convenience sampling was used in the study. The readers need to know the rationale behind that. Please state why you chose to use convenience sampling.
In the data analysis section, the authors stated “Evaluations of the presence of depression, anxiety, and stress were performed according to the scoring system described elsewhere” Can you describe it briefly here? What are your criteria for abnormal levels of depression, anxiety, and stress? The readers need this information to know whether the criteria you used is valid and whether you can compare your research findings with other studies on similar topics.
The authors said “students were categorized based on their GPA into three groups” This is very crucial to all the data analysis. Can you state what specific criteria they used in making the three GPA categories.
In the results section, please make sure you add the GPA categories in the tables.
In the discussion section, when you compare and contrast your research findings with other similar studies, can you summarize the previous studies into cohesive paragraphs instead of just list their research findings individually?
Comments on the Quality of English LanguageThe authors need to go through the manuscript to make some revisions.
Author Response
Comments and Suggestions for Authors
Comment: Thank you for the opportunity to review this manuscript. This is an interesting research topic.
Response: The author of the manuscript appreciates the supportive comment of the reviewer.
Comment: The authors described clearly about the steps they did in this study. But some crucial information needs to be clarified in this study to make it easier for the readers to understand.
Response: The author of the manuscript appreciates the instructive comments of the reviewer. Several modifications were applied to the revised manuscript to enhance the readability of the manuscript.
Comment: The researchers divided the sample according to the GPA category, but it is not clear what specific criteria they used in making the three GPA categories. In several tables (table 1, table 2, table 4) the researchers did analysis according to GPA categories, but the GPA categories were not shown anywhere in the tables.
Response: The criteria used to classify the GPA category is now illustrated in the methodology section of the revised manuscript and highlighted with yellow as the following:
‘Additionally, students were categorized based on their GPA into three groups: those with a GPA of “good” or lower, “very good,” and “excellent.” This classification was made according to the university grading system where students who reported a GPA of 4.5 / 5 or more were classified has having an excellent GPA, students who reported having a GPA ranging between 3.75 - < 4.5 / 5 were classified has having a very good GPA, and those with a GPA lower than 3.75 / 5 were classified as students with a good or lower GPA.’
Comment: The researchers categorize the depression, anxiety, and stress level as abnormal and normal levels, but they did not state the specific criteria they used to make these categorizations. These are very crucial information and the readers need to have this information to understand the study appropriately.
Response: Specific criteria used to categorize the depression, anxiety, and stress levels as abnormal and normal are now detailed and highlighted with yellow in the methodology section of the revised manuscript as the following:
‘Students were classified as either having normal or abnormal levels of anxiety, depression, and stress. Students who had a final score of 9 or less within the depression scale were classified as normal while those who scored more than 9 were classified has having abnormal depression level. Students who had a final score of 7 or less within the anxiety scale were classified has having normal levels of anxiety while those who scored more than 7 were labeled has having abnormal levels of anxiety. Stress final score of 14 or less was classified as normal levels of stress and students who had a final score of 15 or more were classified has having abnormal stress levels.’
Comment: Here is some more feedback on specific paragraphs in different sections. In the introduction section, the authors used “education system’s impact” in paragraph 2. “Education system’s impact” is very broad. Can you narrow it down and make it more specific what impact factors are you focusing on?
Response: The author of the manuscript agrees with the comment of the author. The ‘education system’ term is now removed and replaced with a narrower and more specific alternative. The term ‘academic environment’ is more suitable for the scope of the current paper and more specific. The modification is highlighted with yellow in the introduction section of the revised manuscript.
Comment: On page 2 the paragraph on the association of academic environment and students’ mental well-being is confusing. The topic sentence said academic performance was an influencing factor for mental wellbeing. However, the study cited said the mental health problems had a negative impact on the academic performance in that study. They are two different topics. Please make it consistent what you want to state here.
Response: The author of the manuscript agrees with the comment raised by the reviewer. The topic sentence is now revised to illustrate how impaired mental health of students can affect their academic performance which emphasize on the importance of mental well-being among students. The sentence is now revised and highlighted with yellow in the revised manuscript as the following:
‘The importance of assessing mental well-being among university students stems from the notion that impaired mental health can have an impact on the academic performance of students. Furthermore, the academic stress itself is also a predictor of mental distress among the students. For example, in a Japanese study involving a sample of 1,823 university students, it was concluded that students suffering from impaired mental health in the first semester of their studies were at higher risk of poor academic performance during the remaining study period (5). This notion suggests that poor mental health can harm overall academic performance. However, in another Chinese study that involved a sample of 1804 secondary school students, it was concluded that academic stress can lead to mental distress where the occurrence of mental distress was higher among girls in comparison to boys (6). This indicates the complexity of association between academic environment on mental health of students where poor mental well-being can influence academic performance, and academic distress can be considered as a predictor of mental health status.’
Comment: Paragraph 2 in Page 2 mentioned multiple buffering factors based on a previous study, but none of these factors are included in this study. Please make it clear how does this paragraph fit in the introduction section.
Response: The author of the manuscript agrees with the comment of the reviewer. Nonetheless, providing evidence concerning all factors associated with mental well-being among the students is necessary to provide the readers with comprehensive perspectives of factors associated with mental well-being of the students. Description of these factors might indicate that the complexity of the multifactorial interactions influencing mental well-being of the students. Therefore, it is possible to argue that this paragraph fits in the introduction as it gives the reader a transparent illustration of multiple factors influence the condition of interest in addition to the studied potential impact of physical activity. We acknowledge that factors related to emotional intelligence, supportive parents, positive self-image, mental healthcare seeking behavior are important factors that can influence mental health of the students. However, assessment of these factors goes beyond the scope of the current investigation. To acknowledge this as a limitation, a description is added in the limitation section of the revised discussion and highlighted with yellow in the revised manuscript as the following:
‘Additionally, the findings are limited by the impact of other psychological factors which were not investigated in the current investigation, such as emotional intelligence, the presence of supporting parents, positive self-image, and mental healthcare seeking behavior which might require a conducting future research to indicate how physical activity interact with these factors.’
Comment: Terms like “University environment” “lifestyle characteristics” are all very broad terms. Can you use more specific term to make the statement closely related to the research topic of this study?
Response: The author of the manuscript agree with the comment of the reviewer concerning the broad nature of terms such as “University environment” and “lifestyle characteristics”. To ensure using clearer statement more related to the topic of the research and consistence throughout the manuscript, clearer description while using specific terms was added to the revised introduction of the manuscript and highlighted with yellow as the following:
‘The current literature assessing levels of anxiety, depression, and stress among university students indicates that the academic environment, including the mental distress associated with the level of academic performance, can increase the risk of developing these conditions. Furthermore, certain lifestyle characteristics, such as physical activity, can act as buffering factors, enhancing the mental well-being of the students. However, the of levels of anxiety, depression, and stress according to academic performance and physical activity levels are not well understood.’
Comment: In the method section, as DASS 21 is the main assessment the researchers used to collect information on students’ anxiety, depression, and stress, it is necessary to describe DASS 21 questionnaire’s validity briefly in this manuscript.
Response: The following was added to the methodology section of the revised manuscript and highlighted with yellow to briefly describe the validation of the DASS 21 questionnaire:
‘The second component assessed levels of anxiety, depression, and stress using the DASS 21 questionnaire, whose development, validity, reliability and scoring conducted by Antony et al. (14). The assessment of validity, and reliability of the DASS 21 questionnaire by Antony et al. revealed high internal consistency of where Cronbach's Alpha levels for depression, anxiety, and stress were 0.94, 0.87, and 0.91 respectively. Additionally, concurrent validity of the DASS 21 questionnaire was performed via comparing the findings of the DASS 21 to Beck Depression Inventory, Beck Anxiety Inventory, and State-Trait Anxiety Inventory revealing positive correlations varying between 0.51 and 0.85 between the measured parameters. Furthermore, an Arabic version of the DASS 21 questionnaire was utilized in the current investigation to suit the language preferences of the targeted students (15). In addition to the established validity and reliability of the DASS 21 questionnaire in previous research, the estimated of Cronbach's Alpha test for the current data reached 0.95.’
Comment: It is good you make it clear convenience sampling was used in the study. The readers need to know the rationale behind that. Please state why you chose to use convenience sampling.
Response: Convenience sampling was selected in the current investigation since the recruitment of the participants was performed in online settings. A web link was created and advertised on social media platforms associated with university students, such as WhatsApp and Telegram. Therefore, it was difficult to establish a random sampling process since there was no specific sampling frame when utilizing an online approach and a convenience sampling process was established. Explanation of this notion is now added to the methodology section of the revised manuscript and highlighted with yellow as the following:
‘Convenience sampling was used to reach the targeted students. This method of sampling was utilized since it was difficult to establish a sampling frame when targeting the students via social media platforms.’
Comment: In the data analysis section, the authors stated “Evaluations of the presence of depression, anxiety, and stress were performed according to the scoring system described elsewhere” Can you describe it briefly here? What are your criteria for abnormal levels of depression, anxiety, and stress? The readers need this information to know whether the criteria you used is valid and whether you can compare your research findings with other studies on similar topics.
Response: Criteria for abnormal levels of depression, anxiety, and stress are now described in the methodology section of the revised manuscript and highlighted with yellow as the following:
‘Students were classified as either having normal or abnormal levels of anxiety, depression, and stress. Students who had a final score of 9 or less within the depression scale were classified as normal while those who scored more than 9 were classified has having abnormal depression level. Students who had a final score of 7 or less within the anxiety scale were classified has having normal levels of anxiety while those who scored more than 7 were labeled has having abnormal levels of anxiety. Stress final score of 14 or less was classified as normal levels of stress and students who had a final score of 15 or more were classified has having abnormal stress levels.’
Comment: The authors said “students were categorized based on their GPA into three groups” This is very crucial to all the data analysis. Can you state what specific criteria they used in making the three GPA categories.
Response: The criteria used to classify the GPA category is now illustrated in the methodology section of the revised manuscript and highlighted with yellow as the following:
‘Additionally, students were categorized based on their GPA into three groups: those with a GPA of “good” or lower, “very good,” and “excellent.” This classification was made according to the university grading system where students who reported a GPA of 4.5 / 5 or more were classified has having an excellent GPA, students who reported having a GPA ranging between 3.75 - < 4.5 / 5 were classified has having a very good GPA, and those with a GPA lower than 3.75 / 5 were classified as students with a good or lower GPA.’
Comment: In the results section, please make sure you add the GPA categories in the tables.
Response: description of GPA categories is now added beneath tables 1, 2, 4 and 5. The modified tables are highlighted with yellow in the revised manuscript.
Comment: In the discussion section, when you compare and contrast your research findings with other similar studies, can you summarize the previous studies into cohesive paragraphs instead of just list their research findings individually?
Response: The author of the manuscript appreciate the comment of the reviewer. A comprehensive discussion of all previous similar research and comparing it to the current findings was ensured when composing the discussion section of the manuscript. However, it is possible to argue that listing the findings of similar previous research in individual paragraphs allows comparing the findings between the current and previous research and performing critique of any potential similarities or differences. It would be very difficult to compare specific findings detected in our investigation to specific similar notions detected in another similar studies if the discussion was made collectively containing multiple articles with different findings. Therefore, while ensuring utmost clarity when writing the discussion, no modification to the discussion section was performed in response to this particular comment.
Comment: Comments on the Quality of English Language
The authors need to go through the manuscript to make some revisions.
Response: The author of the manuscript appreciate the comment of the reviewer. A language editing service was performed by Proofed, a UK-based professional service, prior to submission. Nonetheless, we agree to perform a language editing service of the revised manuscript while adhering to the resubmission timeline provided by the journal.
Reviewer 2 Report
Comments and Suggestions for Authors
Thank you for your work. A research done with great effort. There are some shortcomings. By eliminating these, research will be in a better position.
Enjoy your work. I wish you success.
Citation of references does not comply with journal writing rules. should be edited.
Introduction
What was the main hypothesis of this study? The hypothesis should be included in the introduction before writing the purpose statement. The purpose of the research should be presented more clearly.
Method
It is stated in the summary that the research was conducted on 506 people. There is no detail regarding this information in the the method section.
Why what was done with 506 people? How was this number determined? Is this sample size sufficient? Does it represent the universe? For this, a power analysis must be done.
What was the cronbcah alpha value of the DASS 21 Scale found for your study? It must be written.
Also, has the normality of the distribution been tested? The statistical analyzes to be applied were decided according to necessity. It should be explained.
Results
Tables do not comply with journal writing rules. should be edited.
Mean and standard deviation values ​​of any findings are not included.
p values ​​should be written clearly.
References
References are not written according to the journal writing rules.
Additionally, a very insufficient number of resources were used. The content needs to be enriched by adding new resources.
Author Response
Comment: Thank you for your work. A research done with great effort. There are some shortcomings. By eliminating these, research will be in a better position.
Enjoy your work. I wish you success.
Response: The author of the manuscript appreciates the supportive comment of the reviewer. Several modifications of the manuscript were applied in response to the reviewer’s comments to enhance its quality.
Comment: Citation of references does not comply with journal writing rules. should be edited.
Response: The author of the manuscript agrees with the comment of the reviewer. A technical error was detected when citing some references and modified accordingly. All references were arranged via Endnote according to the journal referencing template. The modifications were highlighted with yellow in the revised manuscript to allow inspecting the applied edits.
Introduction
Comment: What was the main hypothesis of this study? The hypothesis should be included in the introduction before writing the purpose statement. The purpose of the research should be presented more clearly.
Response: The hypothesis of the current manuscript is illustrated in the last paragraph of the introduction of the revised manuscript and highlighted with yellow as the following:
‘The current literature assessing levels of anxiety, depression, and stress among university students indicates that the academic environment, including the mental distress associated with the level of academic performance, can increase the risk of developing these conditions. Furthermore, certain lifestyle characteristics, such as physical activity, can act as buffering factors, enhancing the mental well-being of the students. However, the of levels of anxiety, depression, and stress according to academic performance and physical activity levels are not well understood.’
Method
Comment: It is stated in the summary that the research was conducted on 506 people. There is no detail regarding this information in the the method section.
Response: Details concerning the recruited sample of 506 students are elaborated in the first paragraph of the results section and highlighted with yellow as the following:
‘A total of 506 students met the inclusion criteria and completed the data collection. Forty-three cases were excluded for not meeting the inclusion criteria. The mean age of the students was 22 (SD: 2.1) years, with 53% being men and 66% coming from health-related colleges.’
Comment: Why what was done with 506 people? How was this number determined? Is this sample size sufficient? Does it represent the universe? For this, a power analysis must be done.
Response: Justification of the recruited sample size is based on a previous study that assessed levels of depression, anxiety, and stress in the same population. The levels of these conditions varied between 39% and 50% in the same population. The estimate of the sample size of the current research was performed via StatCal function of Epi Info software while ensuring it is powered enough to detect these levels of abnormalities within the same community. Description of the estimate is highlighted with yellow in the revised manuscript as the following:
‘A sample size of 500 university students was estimated based on the findings of a previous study that assessed levels of anxiety, depression, and stress among students from the same university (13). This sample size was calculated using the StatCal function of Epi Info software. It was sufficient to detect a prevalence of anxiety, depression, and stress of 50% with a 95% confidence level and a 5% error margin, with an additional 30% increase to account for excluded cases not meeting the inclusion criteria.’
Comment: What was the cronbcah alpha value of the DASS 21 Scale found for your study? It must be written.
Response: Details concerning the established validity and reliability of the DASS 21 scale in previous and current research are now added to the revised methodology of the manuscript and highlighted with yellow as the following:
‘The second component assessed levels of anxiety, depression, and stress using the DASS 21 questionnaire, whose development, validity, reliability and scoring conducted by Antony et al. (14). The assessment of validity, and reliability of the DASS 21 questionnaire by Antony et al. revealed high internal consistency of where Cronbach's Alpha levels for depression, anxiety, and stress were 0.94, 0.87, and 0.91 respectively. Additionally, concurrent validity of the DASS 21 questionnaire was performed via comparing the findings of the DASS 21 to Beck Depression Inventory, Beck Anxiety Inventory, and State-Trait Anxiety Inventory revealing positive correlations varying between 0.51 and 0.85 between the measured parameters. Furthermore, an Arabic version of the DASS 21 questionnaire was utilized in the current investigation to suit the language preferences of the targeted students (15). In addition to the established validity and reliability of the DASS 21 questionnaire in previous research, the estimated of Cronbach's Alpha test for the current data reached 0.95.’
Comment: Also, has the normality of the distribution been tested? The statistical analyzes to be applied were decided according to necessity. It should be explained.
Response: For the purpose of assessing the study hypothesis, all variables analyzed in the current analysis were either binary or categorical data except for the age. Description of statistical analysis of the age variable is now detailed in the methodology section. The modification of the methodology within the revised manuscript was highlighted with yellow as the following:
‘All variables presented in the current analysis were either binary or categorical except for age. Frequencies and proportions were used to summarize binary and categorical variables. Age was initially summarized via mean and standard deviation (SD). Distribution of the age was assessed via visualization of age distribution using histogram and normal distribution curve. To allow performing subsequent inferential analysis, age was categorized as a binary variable using the mean as a cut-off point where the students were classified as either less than 22, or 22 years old or more.’
Results
Comment: Tables do not comply with journal writing rules. should be edited.
Response: The tables are now modified to comply with the journal writing rules.
Comment: Mean and standard deviation values ​​of any findings are not included.
Response: For the purpose of assessing the study hypothesis, all variables analyzed in the current analysis were either binary or categorical data except for the age. Mean and standard deviation of the age is now added to the modified results section of the revised manuscript and highlighted with yellow as the following:
‘All variables presented in the current analysis were either binary or categorical except for age. Frequencies and proportions were used to summarize binary and categorical variables. Age was initially summarized via mean and standard deviation (SD). Distribution of the age was assessed via visualization of age distribution using histogram and normal distribution curve. To allow performing subsequent inferential analysis, age was categorized as a binary variable using the mean as a cut-off point where the students were classified as either less than 22, or 22 years old or more.’
Comment: p values ​​should be written clearly.
Response: P values are written in all tables except for table 5 which contains multivariate regression analysis and adding P values of all estimated OR might affect the design and readability of the table. Therefore, level of the statistical significance indicated in table 5 are added beneath table 5 and highlighted with yellow in the revised manuscript.
References
Comment: References are not written according to the journal writing rules.
Response: References are now added via Endnote while ensuring meeting the journal writing rules.
Comment: Additionally, a very insufficient number of resources were used. The content needs to be enriched by adding new resources.
Response: We confirm that the manuscript is now enriched by several resources which are now added to various sections of the revised manuscript.
Reviewer 3 Report
Comments and Suggestions for Authors
Dear Authors,
I have gone through your manuscript entitled "Protective effect of exercise against depression, anxiety, and stress among university students based on their level of academic achievement." The topic is so relevant and timely, considering the increasing emphasis on mental health in academic circles. Yours is an important problem to study, and it could contribute significantly to the literature.
It needs several revisions along major scientific rigor lines for clarity and quality. My specific, enumerated suggestions follow after the general comments below:
1. Abstract
An abstract should start with a little background to set the scene but then provide an explicit statement of objectives. Results are summarised in precise statistics, including p-values and confidence intervals where appropriate.
The conclusion should summarise major findings and implications of the findings. Statements of limitations and the direction of future research are also included.
2. Introduction
Literature Review: It needs to be extended to very recent studies conducted on the relationship between exercise, mental health, and academic achievement. This would create a sound theoretical basis for your hypothesis.
Aims/ Objectives- The aims of the study must be adequately defined. It is in this instance that any consideration can be given to stating the particular objectives in bullet points.
Authors report: "In addition to the influence of personal characteristics and the academic environment on students’ mental well-being, academic performance has also been suggested as an influencing factor." Authors should add more information about personal characteristics and their impact on depression, I suggesto to cite the following novel original article:
https://doi.org/10.1007/s00404-013-2720-4
https://doi.org/10.1007/s00404-023-07344-7
https://doi.org/10.1111/jog.13728
3. Material and Methods
Inclusion and exclusion criteria: Specify what constitutes inclusion and exclusion. Also, specify why they have been selected. For example, if one is excluding students who never experienced any stressful incident in the last month, say why such a decision was arrived upon.
Measurements: More details concerning psychometric properties for the instruments that will measure depression, anxiety, stress, and academic achievement should be incorporated into the manuscript. For those measures validated within similar populations, citation of relevant studies is warranted.
Statistical Analysis: Clearly indicate the statistical methods used. Include in the text how missing data were handled and which tests were used for determining the normality of the distribution. When appropriate, consider adding multivariate analysis in order to take advantage of the potential confounders.
4. Results Measures of Central Tendency and Dispersion: Below is a very broad table in descriptive statistics for all the variables.
Inferential Statistics: Where possible, effects sizes should accompany the p-value to enable interpretation of the magnitude of effects found.
5. Discussion Interpretation of results: Relate findings to the research question and hypothesis. Discuss how the results provide support, or nullify other studies. If unexpected - suggest alternatives for unexpected findings.
Strengths and Limitations: Highlight the strengths of the current study and its limitations. Describe any possible biases, one of the most common being self-reporting, and how they could have influenced the results.
I hope these recommendations will strengthen this manuscript in many ways.
Best regards,
Comments on the Quality of English LanguageModerate editing of English language required.
Author Response
Comment: I have gone through your manuscript entitled "Protective effect of exercise against depression, anxiety, and stress among university students based on their level of academic achievement." The topic is so relevant and timely, considering the increasing emphasis on mental health in academic circles. Yours is an important problem to study, and it could contribute significantly to the literature.
Response: The author of the manuscript appreciates the supportive comment of the reviewer.
Comment: It needs several revisions along major scientific rigor lines for clarity and quality. My specific, enumerated suggestions follow after the general comments below:
Response: The author appreciates the several constructive comments made by the reviewer. The manuscript is now revised to enhance its clarity and quality. All modifications applied to the revised manuscript are highlight with yellow to allow easier assessment of the modifications.
- Abstract
Comment: An abstract should start with a little background to set the scene but then provide an explicit statement of objectives. Results are summarised in precise statistics, including p-values and confidence intervals where appropriate.
Response: The author appreciates the comment of the reviewer. the abstract is limited by the word count as per the journal regulations. However, an effort was made to introduce a little background, state the objectives, summarize the results with the relevant p values. However, it was very difficult to include several ORs and 95% CI due to the complexity of the analysis and a brief description of most important findings was presented in the abstract section of the manuscript while ensuring full detailing of the results within the manuscript.
Comment: The conclusion should summarise major findings and implications of the findings. Statements of limitations and the direction of future research are also included.
Response: The conclusion briefly describes the most important findings and implications for practice in academic environment. However, due to the limit of the abstract word count, limitations are not elaborated in the abstract but in the discussion section of the revised manuscript.
- Introduction
Comment: Literature Review: It needs to be extended to very recent studies conducted on the relationship between exercise, mental health, and academic achievement. This would create a sound theoretical basis for your hypothesis.
Response: The author of the manuscript appreciates the comment of the reviewer. More emphasis was made to ensure thorough illustration of associations between academic environment and mental health. However, studies that assessed influence of exercise on mental health of university students are limited which indicates that this is an importnat area for investigation.
Comment: Aims/ Objectives- The aims of the study must be adequately defined. It is in this instance that any consideration can be given to stating the particular objectives in bullet points.
Response: The aims of the current manuscript are described and highlighted with yellow in the revised introduction of the manuscript as the following:
‘The current literature assessing levels of anxiety, depression, and stress among university students indicates that the academic environment, including the mental distress associated with the level of academic performance, can increase the risk of developing these conditions. Furthermore, certain lifestyle characteristics, such as physical activity, can act as buffering factors, enhancing the mental well-being of the students. However, the of levels of anxiety, depression, and stress according to academic performance and physical activity levels are not well understood. The current investigation aims to assess the association between academic performance and the risk of anxiety, depression, and stress among university students from Saudi Arabia. Furthermore, an analysis of how physical activity might buffer levels of anxiety, depression, and stress according to the level of academic performance is also carried out.’
However, using bullet points was avoided to ensure meeting the writing style of the journal.
Comment: Authors report: "In addition to the influence of personal characteristics and the academic environment on students’ mental well-being, academic performance has also been suggested as an influencing factor." Authors should add more information about personal characteristics and their impact on depression, I suggesto to cite the following novel original article:
https://doi.org/10.1007/s00404-013-2720-4
https://doi.org/10.1007/s00404-023-07344-7
https://doi.org/10.1111/jog.13728
Response: The author appreciate the comment of the reviewer. However, upon inspecting the indicated novel articles, it was noted that they are discussing mental health associated with pregnancy which is not aligned with the current scope of the current submission. More related references are now cited to add more information about how personal characteristics can impact well-being. The modified section of the introduction is highlighted with yellow as the following:
‘Worsening mental health among university students has been indicated to be reduced by several buffering factors. In a systematic review involving 31 studies that assessed factors associated with mental health among UK university students, it was concluded that psychological strength characteristics, such as coping strategies and emotional intelligence, supportive parents, positive self-image, and the ability to recognize the development of mental health problems and seek healthcare, were associated with a reduced risk of mental health problems among the students (7). Similarly, in a recent study that assessed mental well-being among university students, it was conceptualized that perceived stress induced by academic, financial, and family pressures is modified by factors related by students’ characteristics such as loneliness, self-esteem, personality and coping strategies (8).Furthermore, personality traits have also been indicated as factors affecting mental well-being of students (9, 10).
- Material and Methods
Comment: Inclusion and exclusion criteria: Specify what constitutes inclusion and exclusion. Also, specify why they have been selected. For example, if one is excluding students who never experienced any stressful incident in the last month, say why such a decision was arrived upon.
Response: Students who never experienced any stressful incidents in the last month during the study were not excluded. This was performed to ensure that recruited students has a wide spectrum of mental health levels. Inclusion and exclusion criteria are described in the methodology section as the following:
‘Students who were registered at the university at the time of recruitment were included. Individuals who were graduates of the university or not registered at the university during the recruitment period were excluded.’
Comment: Measurements: More details concerning psychometric properties for the instruments that will measure depression, anxiety, stress, and academic achievement should be incorporated into the manuscript. For those measures validated within similar populations, citation of relevant studies is warranted.
Response: Full description of validity, reliability and scoring of the instruments is now added to the revised section of the methodology and highlighted with yellow as the following:
‘The second component assessed levels of anxiety, depression, and stress using the DASS 21 questionnaire, whose development, validity, reliability and scoring conducted by Antony et al. (14). The assessment of validity, and reliability of the DASS 21 questionnaire by Antony et al. revealed high internal consistency of where Cronbach's Alpha levels for depression, anxiety, and stress were 0.94, 0.87, and 0.91 respectively. Additionally, concurrent validity of the DASS 21 questionnaire was performed via comparing the findings of the DASS 21 to Beck Depression Inventory, Beck Anxiety Inventory, and State-Trait Anxiety Inventory revealing positive correlations varying between 0.51 and 0.85 between the measured parameters. Furthermore, an Arabic version of the DASS 21 questionnaire was utilized in the current investigation to suit the language preferences of the targeted students (15). In addition to the established validity and reliability of the DASS 21 questionnaire in previous research, the estimated of Cronbach's Alpha test for the current data reached 0.95.
Evaluations of the presence of depression, anxiety, and stress were performed according to the scoring system described elsewhere (14). Students were classified as either having normal or abnormal levels of anxiety, depression, and stress. Students who had a final score of 9 or less within the depression scale were classified as normal while those who scored more than 9 were classified has having abnormal depression level. Students who had a final score of 7 or less within the anxiety scale were classified has having normal levels of anxiety while those who scored more than 7 were labeled has having abnormal levels of anxiety. Stress final score of 14 or less was classified as normal levels of stress and students who had a final score of 15 or more were classified has having abnormal stress levels.’
Comment: Statistical Analysis: Clearly indicate the statistical methods used.
Response: Statically analysis, including scoring, classifications, descriptive and inferential analysis are now detailed in the revised manuscript as the following:
‘Data were analyzed using the Statistical Package for the Social Sciences software (version 25). All variables presented in the current analysis were either binary or categorical except for age. Frequencies and proportions were used to summarize binary and categorical variables. Age was initially summarized via mean and standard deviation (SD). Distribution of the age was assessed via visualization of age distribution using histogram and normal distribution curve. To allow performing subsequent inferential analysis, age was categorized as a binary variable using the mean as a cut-off point where the students were classified as either less than 22, or 22 years old or more.
Evaluations of the presence of depression, anxiety, and stress were performed according to the scoring system described elsewhere (14). Students were classified as either having normal or abnormal levels of anxiety, depression, and stress. Students who had a final score of 9 or less within the depression scale were classified as normal while those who scored more than 9 were classified has having abnormal depression level. Students who had a final score of 7 or less within the anxiety scale were classified has having normal levels of anxiety while those who scored more than 7 were labeled has having abnormal levels of anxiety. Stress final score of 14 or less was classified as normal levels of stress and students who had a final score of 15 or more were classified has having abnormal stress levels.
Additionally, students were categorized based on their GPA into three groups: those with a GPA of “good” or lower, “very good,” and “excellent.” This classification was made according to the university grading system where students who reported a GPA of 4.5 / 5 or more were classified has having an excellent GPA, students who reported having a GPA ranging between 3.75 - < 4.5 / 5 were classified has having a very good GPA, and those with a GPA lower than 3.75 / 5 were classified as students with a good or lower GPA. Finally, the level of exercise was classified based on the number of days per week spent performing any type of physical activity. Due to the low levels of physical activity in the recruited sample, students were grouped into either those who exercised at least once per week or those who did not exercise at all during the week.
Inferential statistics were performed in two steps. First, the chi-squared test was used to assess statistical differences in the demographics of the sample, their exercise frequency, and levels of anxiety, depression, and stress according to their GPA category and exercise frequency. Second, to investigate the influence of exercise frequency on the presence of anxiety, depression, and stress, univariate logistic regression was conducted, with the sample divided into three groups based on GPA category. This was followed by multivariate logistic regression to account for the possible confounding effects of other demographic factors on the association between exercise frequency and levels of anxiety, depression, and stress. A P-value of 0.05 or less was considered statistically significant.’
Comment: Include in the text how missing data were handled
Response: Since data collected in this investigation are primary data, there was a limited impact of missing data and all data presented in the current investigation were complete. Furthermore, cases which were not meeting in inclusion criteria were excluded where the number of excluded cases was declared early in the first paragraph of the results section.
Comment: and which tests were used for determining the normality of the distribution.
Response: For the purpose of assessing the study hypothesis, all variables analyzed in the current analysis were either binary or categorical data except for the age. The distribution of age was initially performed via estimation of mean and strand deviation and visualization via histogram and normal distribution curve. The modification of the methodology within the revised manuscript was highlighted with yellow as the following:
‘Data were analyzed using the Statistical Package for the Social Sciences software (version 25). All variables presented in the current analysis were either binary or categorical except for age. Frequencies and proportions were used to summarize binary and categorical variables. Age was initially summarized via mean and standard deviation (SD). Distribution of the age was assessed via visualization of age distribution using histogram and normal distribution curve. To allow performing subsequent inferential analysis, age was categorized as a binary variable using the mean as a cut-off point where the students were classified as either less than 22, or 22 years old or more.’
Comment: When appropriate, consider adding multivariate analysis in order to take advantage of the potential confounders.
Response: Multivariate analysis was performed as a part of the regression analysis. Section indicating use of multivariate analysis is detailed in the methodology section of the revised manuscript as the following:
‘Second, to investigate the influence of exercise frequency on the presence of anxiety, depression, and stress, univariate logistic regression was conducted, with the sample divided into three groups based on GPA category. This was followed by multivariate logistic regression to account for the possible confounding effects of other demographic factors on the association between exercise frequency and levels of anxiety, depression, and stress. A P-value of 0.05 or less was considered statistically significant.’
Comment: 4. Results Measures of Central Tendency and Dispersion: Below is a very broad table in descriptive statistics for all the variables.
Response: For the purpose of assessing the study hypothesis, all variables analyzed in the current analysis were either binary or categorical data except for the age. The distribution of age was initially performed via estimation of mean and strand deviation and visualization via histogram and normal distribution curve. The modification of the methodology within the revised manuscript was highlighted with yellow as the following:
‘Data were analyzed using the Statistical Package for the Social Sciences software (version 25). All variables presented in the current analysis were either binary or categorical except for age. Frequencies and proportions were used to summarize binary and categorical variables. Age was initially summarized via mean and standard deviation (SD). Distribution of the age was assessed via visualization of age distribution using histogram and normal distribution curve. To allow performing subsequent inferential analysis, age was categorized as a binary variable using the mean as a cut-off point where the students were classified as either less than 22, or 22 years old or more.’
Comment: Inferential Statistics: Where possible, effects sizes should accompany the p-value to enable interpretation of the magnitude of effects found.
Response: The author of the manuscript confirm that p values, effect size, with relevant confidence intervals were used in the tables, especially table 5.
Comment: 5. Discussion Interpretation of results: Relate findings to the research question and hypothesis. Discuss how the results provide support, or nullify other studies. If unexpected - suggest alternatives for unexpected findings.
Response: The author of the manuscript confirms that the results of the current study are compared to similar relevant literature where comparisons were made to discuss how the findings of the current study were either similar or different with justification of the detected differences.
Comment: Strengths and Limitations: Highlight the strengths of the current study and its limitations. Describe any possible biases, one of the most common being self-reporting, and how they could have influenced the results.
Response: To ensure highest level of scientific transparency, limitations of the current study as described in the revised manuscript and highlighted with yellow as the following:
‘However, a major limitation of the study is the low level of physical activity reported among participants, which restricted the ability to further analyze the impact of different types and intensities of physical activity on mental well-being. This limitation is consistent with existing literature indicating low physical activity levels among Saudis, particularly among female students (25). Additionally, the findings are limited by the impact of other psychological factors which were not investigated in the current investigation, such as emotional intelligence, the presence of supporting parents, positive self-image, and mental healthcare seeking behavior which might require a conducting future research to indicate how physical activity interact with these factors. Finally, this study utilized a cross-sectional design to assess the association and assessment of temporality is not possible and it is not clear whether mental distress lead to an impact on academic performance or whether the academic performance lead to the mental distress. However, the findings of the current study, along with relevant literature on factors influencing mental well-being among university students, underscore the importance of implementing organizational programs for the early detection of at-risk individuals and the development of suitable preventive health interventions based on risk assessments.’
Comment: I hope these recommendations will strengthen this manuscript in many ways.
Best regards,
Response: The author of the manuscript highly appreciates the constructive comments of the reviewer. Several modifications are performed and highlighted with yellow in the revised manuscript. We are confident that the comments have strength the reporting quality of the manuscript.
Comment: Comments on the Quality of English Language
Moderate editing of English language required.
Response: The author of the manuscript appreciate the comment of the reviewer. A language editing service was performed by Proofed, a UK-based professional service, prior to submission. Nonetheless, we agree to perform a language editing service of the revised manuscript while adhering to the resubmission timeline provided by the journal.
Round 2
Reviewer 2 Report
Comments and Suggestions for Authors
It appears that many requested corrections have been made.
I wish you success in your future researches.